# Competency analysis based on accounting career anchors using clustering techniques

**Jorge Sánchez-Garcés**[1], **Nelly Rosario Moreno-Leyva**[1], **Lorena Marténez Soto**[2], **Alex Danny Chambi-Rodriguez**[3], **Dina Milagros Tapara-Yanarico**[1], **Dennis Karlo Silva-Vargas**[1], **Himer Avila-George**[4]*

1 Escuela Profesional de Contabilidad, Facultad de Ciencias Empresariales, Universidad Peruana Unién, Puno, Peré, 2 Facultad de Ciencias Administrativas y Contables, Corporación Universitaria Adventista, Medellén, Colombia, 3 Escuela Profesional de Ingenieréa de Industrias Alimentarias, Facultad de Ingenieréa y Arquitectura, Universidad Peruana Unién, Puno, Peré, 4 Departamento de Ciencias Computacionales e Ingenieréas, Universidad de Guadalajara, Ameca, Jalisco, México

* himer.avila@academicos.udg.mx

**Data Availability Statement:** https://github.com/jasg1612/anchors.

## Abstract

This research work aims to identify the prevalent anchors in the professional accounting career using the Schein scale and to describe the prevalent anchors by defining the values, attitudes, aptitudes, skills, and interests. Career anchors are defined by the competence, motivation, and values a person has to perform a particular job in an organization and are present throughout their working life. When determining the soft and hard competencies of the professional profile, universities must consider the career anchors essential for graduates' work performance. To determine which anchors dominate the competencies of the graduate profile, two universities in Latin America with a degree in accounting were selected. The study was organized in two stages: first, the operationalization of the research was conducted, including the description of the instrument through the application of 40 questions divided into Schein's eight anchors. Samples were selected based on the convenience of the authors: one university in Peru and another in Colombia. The sample includes all students enrolled in the accounting major, and the data were coded and processed. In the second stage, data analysis was performed by grouping parameters, analysis of variance, explanatory analysis using a test for the best clustering algorithm, statistical testing, and discussion of the findings. The predominant anchors in the two universities are creativity, entrepreneurship, and lifestyle. The selected universities placed considerable emphasis on training future accountants with an innovative spirit, integrity, and social commitment without neglecting the professional requirements. This approach allows students to undertake challenges and new businesses in their field of work.

## Introduction

Lambert et al. [1] mention the dynamics of professions due to changes in market demands and technological innovations. Therefore, it is necessary that academic institutions take measures to ensure that the training of the professional has some coherence with the changes and

**Funding:** The authors received no specific funding for this work.

**Competing interests:** The authors have declared that no competing interests exist.

requirements of the labor market, seeking good job performance. On the other hand, the labor field has a diversity of requirements; for example, Sheveleva and Pankratova [2] mention that there are requirements focused on the very technical part of the profession, others focused on the service and contribution to the community, others focused on the entrepreneurship of new products, ideas, and innovations. In this sense, it is important to consider forming professional skills related to the mentioned requirements. Then, these requirements become a series of factors closely related to the values and business objectives that make the formation integral. It considers the technical knowledge of the professional career and soft skills such as leadership, creativity, and service, avoiding that students prioritize economic benefits instead of those mentioned above.

Schein [3] is one of the authors who describes these factors in the integral formation, considering the part of knowledge, technical, service, entrepreneurship, creativity, and motivation in challenges, among others. He called these factors career anchors, which are the skills, motivations, and values people develop to consolidate a professional profile. These factors are closely related to the objectives and values of organizations. According to Schein, there are eight types of anchors: technical-functional, management, autonomy-dependence, security-stability, entrepreneurial creativity, service-dedication, challenge (pure challenge), and lifestyle. These anchors are present throughout working life. In the same way, Brooks [4], basing his research on Schein's studies, notes that everyone has a dominant career anchor that will guide the person's current and future roles. We must know what our career anchor is to know how we want to be managed and rewarded within an organization. The details of both proposals are shown in Table 1.

A review of the current state of the research field found that most recent publications support the validity of the career anchors model proposed by Schein. Career anchors remain relevant in conducting studies linked to a wide diversity of objectives, methodologies, and variables; therefore, the proposal is based on the data shown in Table 2.

The limitations found in the literature were potential participant selection biases, small samples, and representation of diverse demographic groups [9, 10]. This research work has the advantage of addressing an occupation (accounting students) outside the organizational setting and focusing on the academic context. It studies how future accountants perform in the

**Table 1. Career anchors.**

| Career anchors | Schein [3] | Brooks [4] |
|---|---|---|
| Technician/Functional | This anchor represents people who possess a great talent for something in particular and focus their motivation on exercising it. | It is the application of an individual's skills in a similar area in order to perfect them. |
| General management | These are people whose interest is climbing the corporate ladder to high levels of responsibility and leadership. | This type of person seeks a very high level in an organization because of the responsibility. |
| Autonomy/ Independency | Describes people who consider organizational life restrictive and do not seek to perform under rules, procedures, and other standards imposed by others. | Conceptualizes how an individual defines a job by his or her criteria. |
| Security/Stability | Describes individuals who base their decisions on their financial security and job security. | It considers those individuals who are subject to job security above all else. |
| Creativity in entrepreneurship | It identifies dreamers and individuals with ingenuity for creation or innovation in the business world. | They possess confidence in their ability to create new organizations by taking risks and overcoming obstacles. |
| Services/Dedication | They are individuals whose work decisions are based on their values rather than their real talents; they want to improve the world. | Such individuals are characterized by a desire to make the world a better place to live; they seek harmony of the general welfare. |
| Challenges | They are people whose definition of success is overcoming impossible obstacles. | They are those who like to work in conflict spaces. |
| Lifestyle | Describes people who possess an organizational attitude that reflects respect for personal and work concerns. | Consider those individuals who seek a balance between personal needs and those of their family. |

**Table 2. Representative publications that address the issue of career anchors from different solution approaches.**

| Proposal | Techniques | Results | Ref. |
|---|---|---|---|
| Analyzing occupational change through 10-year longitudinal data analysis | Simple linear regression to predict occupational change | Change factors transcending occupational mobility. | [5] |
| Defines a proposition based on orientation. Based on preferences arising from the interaction between self-identification, family relationships, social and cultural background, education, work experience, and labor market conditions. | Qualitative categorical analysis with Nvivo software about what was valued most at work and main career goals. | Competencies were categorized into fifteen categories representing possible orientations. | [6] |
| Analyze the integration of contemporary career orientation concepts with career self-management. | Bibliographic review. | Model career orientations as antecedents of career self-management behaviors and career outcomes. | [7] |
| Demonstrate that networking allows for strengthening work skills, gaining opportunities such as relating and working in diverse contexts, and continuous learning. | Semistructured interviews with an exploration of responses and coding. | Professional development networks are used for a purpose and influence the assignment of the most interesting projects and the improvement of professional skills through exchanges of experience. | [8] |
| To determine whether, according to the career anchors model proposed by Schein, the expectations, strategies, and experiences of accounting-finance professionals match the new requirements of the labor market. | A research strategy categorized as a three-phase mixed methods study with an exploratory sequential approach was used. | There are new perceptions of the importance of career anchors associated with the acquisition of competencies that help maintain the individual employability of accountants. | [1] |
| To identify the predominant career anchors of Japanese occupational health nurses. | The study used a descriptive qualitative approach because it was considered the most appropriate for describing the work of occupational health nurses and the influence of work on their private lives. | The results showed that the most important skills in occupational health nurses in Japan were relationship and position management at work, the ability to execute occupational health practices, and management skills for effective work, among others. | [9] |
| To determine the predominant career anchors of graduate management students in India to identify possible variations among different majors of study. | Schein's career anchors scale was used, and comparisons between the career anchors of the demographic groups were performed using factor analysis and panel data regression. | Significant differences were found between the predominant career anchors of individuals with different graduate management majors. | [10] |
| To determine how in IT professionals, individual preferences about organizational career management vary according to career anchors. | An online questionnaire was used, in which 1629 professionals from 10 organizations in Switzerland, Germany, and the United Kingdom participated. Igbaria and Baroudi's (1993) instrument was used to measure career anchors. | Connections between career anchor scores and preferences for different types of organizational career management practice were observed. These connections were more evident for some than for others. | [11] |
| To analyze the relationship between gender and career anchors of undergraduate students. | The exploratory and cross-sectional survey method was used with a sample of 251 engineering students and 251 health students. | In both engineering and health areas, the results were more related to the predominant anchors according to the socio-cultural role determined by gender than that of the profession. | [12] |
| To examine the career anchors of a sample of certified public accountants (CPAs) to assess the relationship between their career anchors and their work experiences and attitudes. The study included three industry sectors: public practice, industry, and government. | A sample of 1440 CPAs was considered. The CPAs were randomly selected from those identified as working in public accounting, governmental accounting, and managerial accounting. | The study suggests that when there is alignment between career anchors and work context, there is greater engagement, higher job satisfaction, and lower turnover intentions among CPAs. | [13] |
| To identify the interconnectedness of career orientations and motivations related to the educational and work activities of psychology and pedagogical education students. | A total of 114 individuals, male and female, between 17 and 21 years of age, were surveyed. Spearman's correlation analysis, Friedman's criterion, and the Mann-Whitney criterion were used for data analysis. | The career anchors service/dedication to a cause, pure challenge, and technical challenge/functional competence were the most influential for career exploration. | [2] |

labor field in relation to the inequality between competencies and skills provided by higher education institutions. Therefore, the possibility of impacting the populations studied is meritorious in offering greater clarity on the relationship between the requirements of the labor market and the professional profile.

This research work aims to identify the prevalent anchors in the professional accounting career using the Schein scale and to describe the prevalent anchors by defining the values, attitudes, aptitudes, skills, and interests of accounting students from two South American higher

education institutions. When determining the prevalent anchors of the accounting profile, it would be possible to observe the motivations that lead the professional to stand out in the labor field, achieving an excellent performance, which generates an impact on the valuation of the professional. Therefore, the research responds to the need to have information from different groups of countries, which enables the identification of contexts that do not limit the research. This information includes culture, occupation, and profession, which enriches the classification of the sample. It avoids being limited to a single response categorization, as observed in many studies, such as Schein's.

The remainder of this paper is structured as follows: Section 2 describes the sample used, and the proposed method is presented in detail. Section 3 provides the results grouped by country. Section 4 presents a discussion of the results. Finally, Section 5 presents the conclusions.

## Materials and methods

### Sample

The sample applied in this research is nonprobabilistic, and for the convenience of the authors, the sample consisted of 523 students intending to have a career in accounting. There were 459 respondents of Peruvian nationality and 64 of Colombian nationality. The ages of the Peruvian students ranged from 16 to 57 years, and the ages of the Colombian students ranged from 16 to 33 years.

[14] provides the formula 50+8 (x), where x is the number of independent variables analyzed (8 career anchors); therefore, 50+8 (8) = 114 would be the minimum sample for the inferential analysis. In both samples, the total number of students from both countries in the accounting career was considered, as detailed in Tables 3 and 4.

### Research methodology

The research was organized into two phases, as shown in Fig 1. The first consisted of obtaining and processing the data, and the second consisted of data analysis using tools such as clustering and analysis of variance (ANOVA). The research is of a quantitative explanatory level because clusters that group the data for each sample obtained from each country are identified. The predominant anchors are also described.

**Description of the instrument.** The instrument used in this research was proposed by [15], who elaborated the questionnaire based on the model proposed by Schein [3] and recently reviewed by Brooks [4], where they show that skills such as work and professional

**Table 3. Peru sample.**

| Headquarter | Female | Male | Total |
|---|---|---|---|
| Juliaca | 206 | 112 | 318 |
| Lima | 50 | 37 | 87 |
| Tarapoto | 39 | 15 | 54 |
| Total | 295 | 164 | 459 |

**Table 4. Colombia sample.**

| Headquarter | Female | Male | Total |
|---|---|---|---|
| Principal | 34 | 30 | 64 |

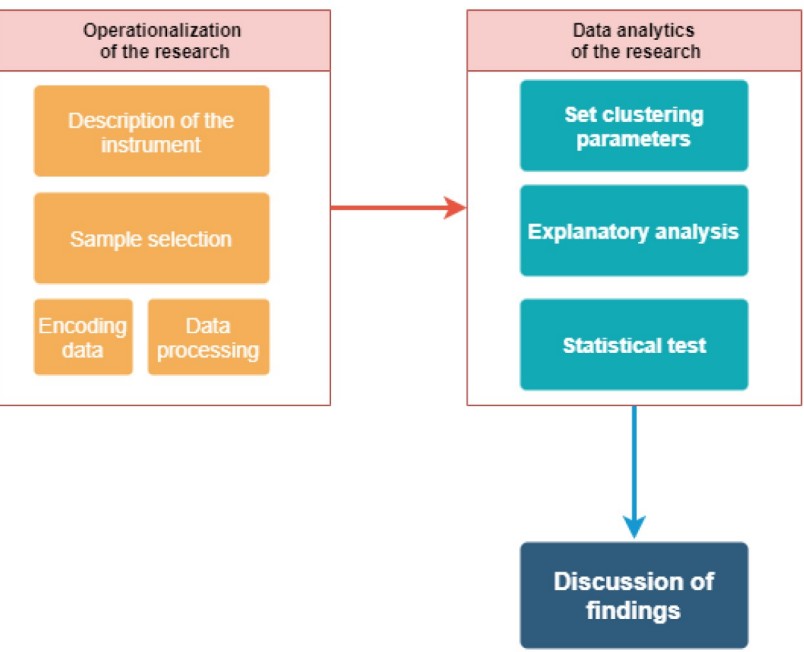

**Fig 1. Description of the proposed methodology.**

performance are part of one of the career anchors. The identification of these career anchors enables individuals to evaluate their preferences and improve their professional performance. Therefore, the instrument proposed by Medina in 2012 consists of 40 questions that are divided into Schein's eight anchors; see Table 5.

**Sample selection.**

- Inclusion: Students enrolled in the accounting career.

- Exclusion: Students who did not study accounting were excluded and students who were not enrolled in the academic term at the time the survey was conducted.

**Encoding data.** The categorical attributes of text type, gender, and campus were coded; see Table 6. For this purpose, the repeated data in the categorical column were discriminated, and once unique categories were obtained, they were converted to numerical data by means of a data dictionary that equates text data to the respective numerical meaning [16].

**Data processing.** The data were processed based on the references of each question concerning its anchor, as explained in Table 5. Therefore, the scores of all questions related to the

**Table 5. Questions related to its anchor.**

| Career anchors | Questions |
|---|---|
| Technician/Functional | 1, 9, 17, 25, 33 |
| General management | 2, 10, 18, 26, 34 |
| Autonomy/Independence | 3, 11, 19, 27, 35 |
| Security/Stability | 4, 12, 20, 28, 36 |
| Creativity in entrepreneurial | 5, 13, 21, 29, 37 |
| Service/Dedication | 6, 14, 22, 30, 38 |
| Challenges | 7, 15, 23, 31, 39 |
| Lifestyle | 8, 16, 24, 32, 40 |

**Table 6. Description of the demographic attributes of the dataset.**

| Attribute | Description |
|---|---|
| Age | Age of the student in the accounting program |
| Gender | Student's gender |
| Campus | Country where the University is located |

anchor were added to determine a quantitative value for each anchor and identify the predominant anchor for each student.

**Set clustering parameters.** A wide range of clustering algorithms exist; most of them have many hyperparameters on which the quality of the clustering partition depends [17]. Thus, the silhouette method was used to optimize the hyperparameters of the clustering algorithms used in this study.

**Silhouette method.** The silhouette method was first proposed by Rousseeuw [18]; it is used to measure the quality of clustering. The silhouette method measures the separation distance between clusters and indicates how close each point of a cluster is to the points of neighboring clusters. The silhouette coefficient is calculated using Eq (1). This distance measure is in the range [-1, 1], where a value close to +1 indicates that the point is well cohesive with its group and poorly cohesive with neighboring groups. If most of the points are high, the pool setting is appropriate. If many points are low or negative, the clustering configuration may have too many clusters.

$$s(i) = \frac{b(i) - w(i)}{max\{w(i), b(i)\}},$$ (1)

where $w(i)$ is the mean distance between point $i$ and other points in the same cluster. $b(i) = \min\{b_{i1}, b_{i2}, \ldots, b_{ik}\}$, where $b_{ij}$ is the mean distance of point $i$ to the points of other clusters.

**Explanatory analysis.** Clustering was used to divide the data set taken from the sample of the eight anchors into subsets called clusters and grouped using geometric distance. Clustering was used to reduce the dimensions of the data; in this case, it was applied to find the most prevalent anchors in each sample from Peru and Colombia (the purpose of the study). This process was validated by the Fisher's F statistic of ANOVA, which measures the correlations between the anchors run with cluster 0, identifying by Fisher's estimated value those with the highest prevalence value in the cluster.

K-means clustering, density-based spatial clustering of applications with noise (DBSCAN), and balanced iterative reducing and clustering using hierarchies (BIRCH) were used. These clustering algorithms were selected because they are well-known and because their clustering approaches differ, i.e., partitioning, density, and hierarchy.

K-means clustering consists of two phases: (1) calculation of the cluster centroids and (2) assignment of the data to the closest cluster. These two phases are performed iteratively until the best centroids, in terms of minimizing the sum of the distances of each cluster object to its center, are identified [19]. Fig 2 details the steps of the K-means algorithm.

DBSCAN is a density-based clustering algorithm that can be used to identify clusters of any shape in a dataset containing noise and outliers. The DBSCAN algorithm is based on the intuitive notion of clusters and noise. The key idea is that the neighborhood of a given radius must contain a minimum number of points for each point in a cluster. The DBSCAN algorithm requires two parameters:

---

**Require:** Data points $D$ and number of clusters $k$.
1: Initialize $k$ centroids randomly.
2: **repeat**
3:     For each data point, compute its Euclidean distance from the centroids.
4:     Identify the nearest centroid of all input data points.
5:     Recalculate the positions of the centroids.
6: **until** The smallest error is obtained.
7: **return** Data points with cluster memberships.

---

**Fig 2. K-means algorithm [20].**

- Epsilon (eps): specifies how close points must be to each other to be considered part of a cluster. If the distance between two points is less than or equal to epsilon, these points are considered neighbors.

- Minimum points (MinPts): the minimum number of points to form a dense region. For example, if we set MinPts to 5, then at least 5 points are required to form a dense region.

In this algorithm, there are three types of data points: core points, border points, and noise points. A point is a core point if it has more than a specified number of MinPts within an eps radius around it. Core points always belong to a dense region. A point is a border point if it has fewer than MinPts within eps, but it is in a core point's neighborhood. A noise point is any point that is not a core point or a border point.

The pseudocode of the DBSCAN algorithm is shown in Fig 3. The algorithm starts with an arbitrary point that has not been visited, and its neighborhood information is retrieved from the eps parameter. If the point contains minimal points within the eps neighborhood, clustering is initiated; otherwise, the point is labeled as noise. This point can later be found within the eps neighborhood of a different point and thus become part of that cluster. The concepts of reachable density and density connection points are important. If a point is found to be a core point, then points within the eps neighborhood are also part of the cluster. Thus, all points found within the eps neighborhood are aggregated, along with their own eps neighborhood, if they are also core points. The above process continues until the density-connected cluster is completely identified. The process is then restarted with a new point that may be part of a new cluster or labeled as noise.

---

**Require:** Data points $D$ and global parameters eps, MinPts.
1: Arbitrarily select point $p$.
2: Retrieve all points that are density-reachable.
3: If $p$ is a core point, a cluster is formed.
4: If $p$ is a border point, no points are density-reachable from $p$, and it visits the next point in the database.
5: Continue the process until all the points have been processed.
6: **return** Data points with cluster memberships.

---

**Fig 3. DBSCAN algorithm [21].**

The BIRCH algorithm uses a tree structure to perform clustering quickly. This numerical structure is similar to a balanced B+ tree. Generally, this tree structure is called a clustering feature tree (CF Tree). Each node of such a tree is composed of several clustering features (CFs). The main process of the BIRCH algorithm involves establishing a CF tree, and it consists of four phases:

1. Loading: All samples are read in sequence to create a CF tree in memory.

2. Tree condensing: The CF tree established in the first step is filtered to remove abnormal CF nodes, which usually contain a few sample points.

3. Global clustering: Other clustering algorithms, such as K-means, are used to cluster all CF tuples to obtain a better CF tree. The main objective of this step is to eliminate the sample reading order.

4. Clustering refinement: The centroids of all CF nodes of the CF tree generated in the third step are used as the initial centroid points to cluster all sample points according to distance.

Notably, the BIRCH algorithm does not need to enter the *k* value of the category number; the number of the last CF tuple is the final *k*. For more details about the BIRCH algorithm, the reader can consult [22].

**Statistical test.** ANOVA is a parametric measure of the variability of two or more sets of an experiment. ANOVA is a statistical test that uses the Fisher distribution test (probability distribution) to measure clustering quality and variance. The ANOVA method tests two hypotheses: $H0 = \mu_1 = \mu_2 \ldots = \mu_k$ and $H1$ there is no equality of means in the measured populations. ANOVA can be a one-way or two-way analysis; we use two-way ANOVA, which is an extension of the first. In two-way ANOVA, the influence of independent variables on a variable is examined. In our case, we consider how the career anchors influence the clusters defined in the computational model to determine if the variance between clusters regarding the professional anchor exists. The two-way Fisher's statistics are calculated according to the following steps [23, 24]:

Step 1: Establish the null hypothesis (H0) and the alternative hypothesis (H1).

Step 2: Find the total $T$ of all observations ($x$) in all samples according to

$$T = \sum x_1 + \sum x_2 + \ldots + \sum x_k.$$

Step 3: Find the value of the correction factor, expressed as

$$T^2/N,$$

where $N$ is the total number of samples, expressed by

$$N = n_1 + n_2 + n_3 + \ldots n_k$$

$$SST = \sum x_1^2 + \sum x_2^2 + \ldots + \sum x_k^2 - T^2/N$$

Step 5: Find the sum of the squares of the deviations between the SSB samples according to

$$SSB = \left[\sum x_1^2 + \sum x_2^2 + \ldots + \sum x_k^2\right] - T^2/N$$

**Table 7. List of hyperparameters for each algorithm.**

| Technique | Hyperparameters | values |
|-----------|-----------------|--------|
| Kmeans | n_clusters | 2,3,4,5,6,7,8,9,10 |
| DBSCAN | eps | {0-4} |
| | min_samples | {2-10} |
| | p | {0-20} |
| Birch | branching_factor | {10-50} |
| | n_clusters | {2-20} |
| | threshold | {0-1} |

Step 6: Calculate the mean squared deviations within the samples to be tested (SSW), according to

$$SSW = SST - SSB$$

Step 7: Find the degrees of freedom ($(v_1 = df_1 = k - 1)$ ($k$ is the number of columns) and the degrees of freedom ($v_2 = df_2 = N - k$).

Step 8: Find the mean squared deviation between samples (MSB), according to

$$MSB = SSB/v_1$$

and the within-sample mean squared deviation (MSW), according to

$$MSW = SSW/v_2$$

Step 9: Calculate the F-statistic by means of Eqs (2) and (3).

$$F = MSB/MSW \rightarrow MSB > MSW \qquad (2)$$

$$F = MSW/MSB \rightarrow MSW > MSB \qquad (3)$$

## Results

### Clustering method evaluation

In this section, the results of applying the methods to obtain the optimal number of clusters and to compare the three clustering algorithms explained in Section Explanatory analysis. The algorithms were tested using hyperparameters according to Table 7 and then compared using the Silhouette score. The results shown in Table 8 indicate that DBSCAN obtained the best score of 0.42 for Peru and 0.32 for Colombia. This algorithm detected a single cluster, which

**Table 8. Comparison of clustering methods: Peru and Colombia.**

| Country | Cluster method | best hyperparameters | Silhouette score |
|---------|----------------|----------------------|------------------|
| Peru | KMEANS | n_clusters = 2 | 0.3 |
| | DBSCAN | eps: 3.4898111793920927, min_samples:3, p: 14 | 0.42 |
| | BIRCH | branching_factor: 19, n_clusters: 2, threshold: 0 | 0.25 |
| Colombia | KMEANS | n_clusters = 2 | 0.21 |
| | DBSCAN | eps: 3.8403836485017817, min_samples:3, p: 1 | 0.32 |
| | BIRCH | branching_factor: 15, n_clusters: 2, threshold: 0 | 0.22 |

**Table 9. Significant anchors for each cluster: Peru and Colombia.**

| Country | Cluster | Age | Gender | TF[1] | GM[2] | Autonomy | Stability | CE[3] | Services | Challenges | Lifestyle |
|---------|---------|-----|--------|-----|-----|----------|-----------|-----|----------|------------|-----------|
| Peru | 0 | 21 | 1 | 21 | 19 | 21 | 21 | 23 | 22 | 22 | 21 |
| Colombia | 0 | 21 | 1 | 21 | 18 | 21 | 22 | 22 | 23 | 21 | 22 |

[1]TF = Technician/Functional

[2]GM = General management

[3]CE = Creativity in entrepreneurship

Gender option 1 = male

Gender option 2 = female

has been labeled 0, and another cluster -1, which represents the data set containing outliers in the samples.

Table 9 shows a summary of the results of the surveys divided by academic institutions located in Peru and Colombia. Columns 3 and 4 describe the respondents' demographic data to contextualize columns 5 to 12, which show the value of each anchor calculated based on the numerical response of the respondents, weighting the anchor according to their capabilities. Then, the responses were summed considering the Table 5 instrument that relates the questions to the anchors; subsequently, each row of the record with its anchor values was labeled with a cluster value. Once the rows were grouped according to the label, they were averaged, obtaining the final value of the prevalent anchor per record row. According to [10], this allows insight into key perspectives and career orientations based on those anchors.

### Peruvian university campus

**Description of cluster significance.**   Once the DBSCAN algorithm labeled the rows of the Peru sample record with each of the clusters (0 and -1), the mean of cluster 0 was calculated. Table 9 summarizes the values of this mean, which is a value of central tendency used in statistics as the balance point of the dataset. For analysis purposes, the mean identified the characteristics of each anchor in the dataset. For example, cluster 0 describes a 21-year-old male student whose most notable professional anchors are service, entrepreneurial creativity, pure challenge, functional technician, autonomy, and stability. The anchor of services related to individuals with a profound service mission. Such individuals feel committed to improving their community's social and economic situation and seek to influence others to transcend in their actions. They are people who like to develop consultancies through independent activities without receiving a means of payment. The entrepreneurial creativity anchor, they have the skills to innovate and create new things, thanks to their well-developed imagination. The pure challenge anchor consists of professionals who like to take risks and participate in challenging competitions. They work in companies where they encounter challenges, such as start-ups, and accept challenges related to new products or services. The functional technician anchor, they take responsibility for detrimental areas in situations that may be considered insurmountable, and their competencies are very high. These individuals are not discouraged and often leave their jobs to seek new challenges. Autonomy represents the need to have freedom of movement to develop your career and have your own style (this is aimed at consultants and small business owners); professionals seeking stability are those who are looking for a secure job without the risk of being fired [15, 25].

Fig 4 describes the level of differences in the cluster groups due to the development and impact that each anchor (professional competence) has on these groups. Each anchor has

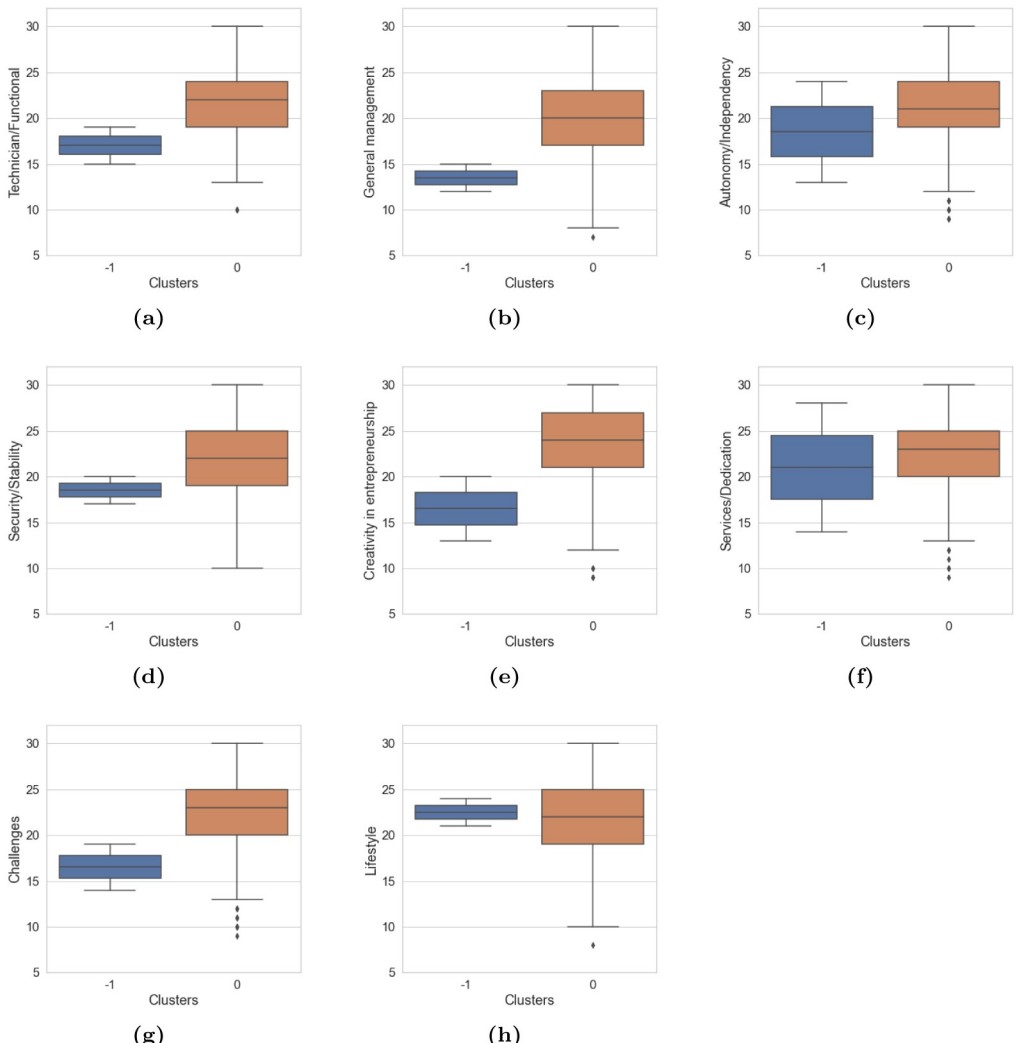

**Fig 4. The behavior of each cluster with respect to the anchor: Peru.** ♦ represents the points outside the quartile range, the blue box describes the cluster sample -1 and the orange box describes the cluster sample 0.

different levels of development and impact on the groups. Therefore, the mean of cluster 0 is higher in all anchors except lifestyle; and the data intervals are larger, i.e., this cluster 0 has a larger number of records and greater representativeness in the analysis. According to the results of the ANOVA analysis shown in Table 10, the anchor with the highest rating was Creativity in entrepreneurship. The factor ($cluster_k \sim Creativity in entrepreneurship$) has the highest Fisher value of 4.93; this indicates that among all the anchors, this anchor has the highest significance in the sample of Peruvian university students; likewise, the Table 9 confirms this value and is corroborated by the p-value of 0.0267, which was the lowest.

## Colombian university campus

**Description of cluster significance.** Table 9 summarizes the mean values of the single cluster 0, and for analysis purposes, the mean identified the characteristics of this cluster in the dataset. The case of the first cluster describes a 21-year-old male junior student whose career

**Table 10. ANOVA table of career anchors: Peru and Colombia.**

| Country | Career anchors | Sum of squares | Degrees of freedom | Fisher | p-value |
|---|---|---|---|---|---|
| Peru | Technician/Functional | 0.010 | 1.0 | 2.84 | 0.0925 |
| | General management | 0.020 | 1.0 | 3.63 | 0.0573 |
| | Autonomy | 0.003 | 1.0 | 0.76 | 0.3826 |
| | Stability | 0.005 | 1.0 | 1.21 | 0.2711 |
| | Creativity in entrepreneurship | 0.020 | 1.0 | 4.93 | 0.0267 |
| | Services | 0.001 | 1.0 | 0.24 | 0.6222 |
| | Challenges | 0.010 | 1.0 | 3.48 | 0.0629 |
| | Lifestyle | 0.000 | 1.0 | 0.04 | 0.8322 |
| Colombia | Technician/Functional | 0.003 | 1.0 | 0.21 | 0.6449 |
| | General management | 0.019 | 1.0 | 1.22 | 0.2728 |
| | Autonomy | 0.020 | 1.0 | 1.40 | 0.2408 |
| | Stability | 0.010 | 1.0 | 1.07 | 0.3060 |
| | Creativity in entrepreneurship | 0.160 | 1.0 | 11.65 | 0.0011 |
| | Services | 0.090 | 1.0 | 6.52 | 0.0131 |
| | Challenges | 0.070 | 1.0 | 5.28 | 0.0249 |
| | Lifestyle | 0.190 | 1.0 | 15.39 | 0.0002 |

anchors are service, stability, lifestyle, entrepreneurial creativity, autonomy, and pure challenge. These characteristics were already mentioned in Section Peruvian university campus.

Fig 5 describes the level of differences that exist in the cluster groups due to the development and impact that each anchor (professional competence) has on these groups. Each anchor has different levels of development and impact on the groups; Therefore, the mean of cluster 0 is higher in all anchors, except in technician / functional; the data intervals are larger; this cluster has a greater number of records and greater representativeness in the analysis. As for cluster -1, only one record was labeled in this cluster, which is illustrated in Fig 5.

According to the ANOVA results in Table 10, the anchor that has the greatest impact on the clusters in terms of obtaining a better segmentation and identification of groups is lifestyle; in this sense, the factor ($cluster_k \sim Lifestyle$) has the highest Fisher value of 15.39. This indicates that among all anchors, this anchor has greater significance in the sample of Colombian university students of both clusters. This finding is corroborated by the p-value of 0.0002, which was the lowest.

## Discussion

Table 11 describes the predominant anchors of the whole sample, i.e., the combined sample of Peruvian and Colombian respondents. To determine this prevalence, two metrics were calculated: F-ANOVA and Test.

The first metric calculated was the F-statistic of the variance. According to [26], a higher F-value indicates a more significant impact on the sample and a more relevant characteristic (career anchor).

The second metric was obtained from the sum of the questions related to each anchor according to Table 5. Each column of the anchor corresponding to the entire sample was added, resulting in the total value per anchor of the entire sample (Peruvians and Colombians).

To define the prevalent career anchors in the whole sample, a column called count was added, where for each minimum metric, values to be met were considered according to the

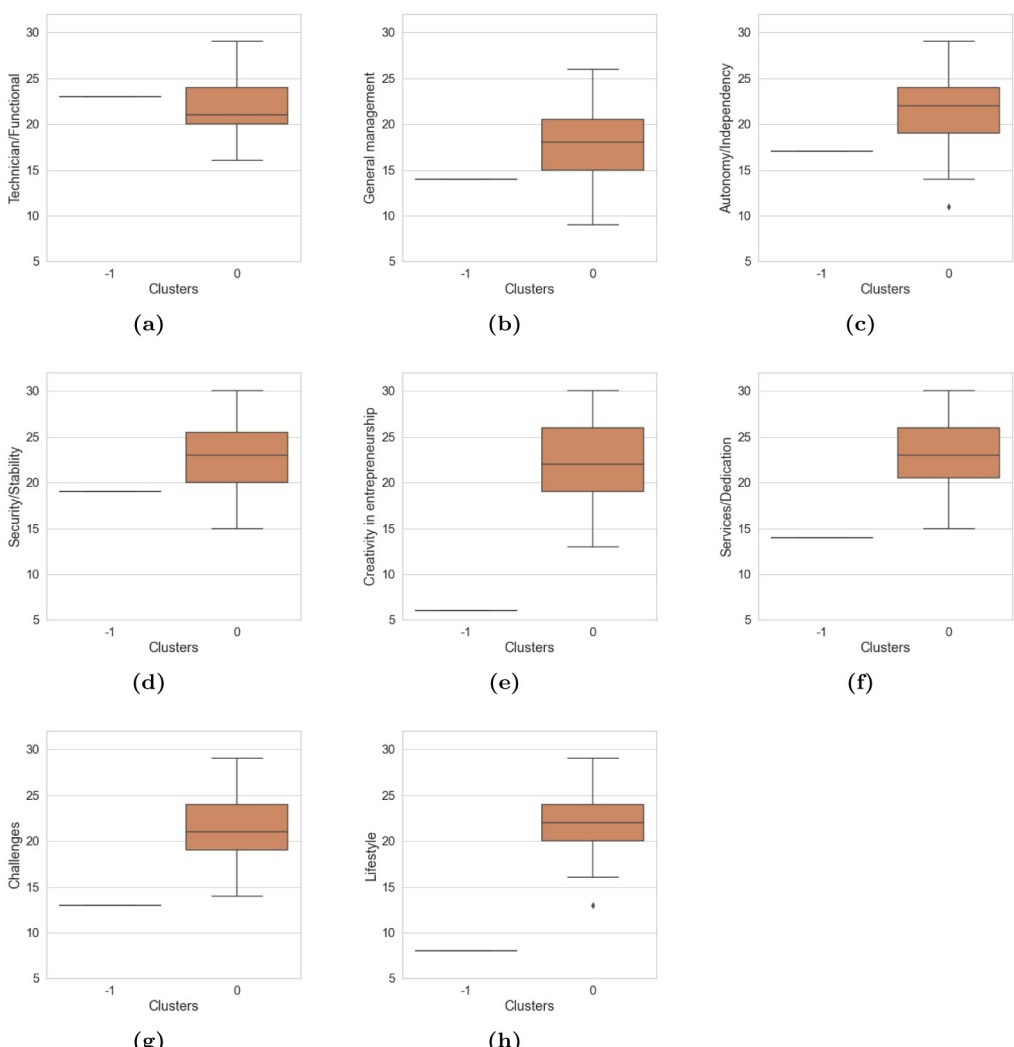

**Fig 5. The behavior of each cluster with respect to the anchor: Colombia.** ♦ represents the points outside the quartile range, the blue box describes the cluster sample -1 and the orange box describes the cluster sample 0.

**Table 11. Predominant anchors of the whole sample.**

| Anchors | F-ANOVA | Test | Count |
|---|---|---|---|
| Technician/Functional | 443 | 9894 | 1 |
| General management | 348 | 9061 | 0 |
| Autonomy | 399 | 9712 | 0 |
| Stability | 362 | 9955 | 0 |
| Creativity in entrepreneurship | 290 | 10733 | 1 |
| Services | 370 | 10315 | 1 |
| Challenges* | 436 | 10201 | 2 |
| Lifestyle | 398 | 10046 | 1 |

* Indicates the most representative anchors according to the count column.

maximum score obtained in the metric. For ANOVA, the minimum value was 400, and the test was 10000; in this sense, the predominant anchor was Pure Challenge. The explanation of this anchor is found in section Peruvian university campus.

Ona [27] conducted a study with 437 students of the Technical University "Gheorghe Asachi," Romania. He used an independent sample t-test to examine how the values that determine career aspirations differ. Among the relevant anchors are Pure Challenge and Service due to the need to overcome any obstacle. Notably, the combination of both anchors indicates that individuals like to address complicated tasks and difficult situations without losing sight of the need for change, a vision of service, and influence in acting according to professional and ethical values. These events can be evidence of maturity and a better understanding of the available labor market and economic environment. Similarly, Weber and Ldkin [28], in a study of 693 industry professionals from countries such as Hong Kong, Singapore, Thailand, and Malaysia, showed that the pure challenge career anchor is closely related to professional identity and, therefore, to corporate identity and belonging. Such individuals take on challenges with the commitment to achieve change and improvement, demonstrating the ideals and importance of the professional career to which they belong. The results of the study have implications for management, and suggestions for future research are offered.

Additionally, Demel and Mayrhofer [25], in a qualitative study with semistructured interviews with 40 internationally mobile Austrian professionals working in different European countries, note that what is important for these individuals are the combination of the anchor's entrepreneurial creativity, pure challenge, service, professional technician. These anchors allow them to shape something new, exist in conditions of constant innovation and change, influence others when making very complex decisions, be sure that the contribution to the process of change will be very good, be open to learning based on international experience and expand networks of contacts at an international level. In this sense, the authors mention the importance of the study of anchors, especially in the development of professional skills and attitudes for good job performance.

## Conclusion

The skills performed and attitudes in the working world are is of vital importance for choosing a career, forming a vision and purpose for life, and having a sense of work and roles fulfilled. The anchors creativity and entrepreneurship obtained high scores in cluster 0 of Peru and Colombia, and the ANOVA table of Peru and lifestyle shows high scores in cluster 0 of Colombia, the ANOVA of Colombia. Both anchors refer to professionals who need to innovate and create new things. However, to achieve this purpose, the lifestyle anchor, which applies to individuals who want their work to adapt to their needs (schedules, geographic location, etc.), is essential for people who need to innovate and create new things. There is likely a strong influence of culture in each country, which contributes to the profile of the public accountant. In addition, each university institution promotes a series of specific characteristics within its institutional profile that could influence how students interpret their professional careers. Therefore, these profiles have a greater orientation to service, innovation, professional ethical values, and strengthening of professional technical skills, promoting the stability of future professionals and motivating them to be more focused on finding a job that will allow them to enter the labor market as employees in an organization that provides greater stability and less risk.

Future work will expand the sample to other cultural contexts and origins of other countries in South America, broadening the context of the profile of the public accountant and

highlighting the most relevant anchors of these countries to formulate a potential profile of professionals in this part of the American continent.

## Author Contributions

**Conceptualization:** Jorge Sánchez-Garcés, Nelly Rosario Moreno-Leyva, Lorena Marténez Soto.

**Data curation:** Alex Danny Chambi-Rodriguez, Dina Milagros Tapara-Yanarico, Dennis Karlo Silva-Vargas.

**Formal analysis:** Jorge Sánchez-Garcés, Alex Danny Chambi-Rodriguez, Himer Avila-George.

**Investigation:** Nelly Rosario Moreno-Leyva, Lorena Marténez Soto, Himer Avila-George.

**Methodology:** Nelly Rosario Moreno-Leyva, Dina Milagros Tapara-Yanarico, Dennis Karlo Silva-Vargas, Himer Avila-George.

**Validation:** Lorena Marténez Soto, Alex Danny Chambi-Rodriguez, Dina Milagros Tapara-Yanarico, Dennis Karlo Silva-Vargas, Himer Avila-George.

**Visualization:** Jorge Sánchez-Garcés, Himer Avila-George.

**Writing – original draft:** Jorge Sánchez-Garcés, Nelly Rosario Moreno-Leyva, Lorena Marténez Soto, Alex Danny Chambi-Rodriguez, Himer Avila-George.

**Writing – review & editing:** Himer Avila-George.

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
