## [Decision Letter · Decision Letter 0]

28 Feb 2022

PONE-D-22-01029Competency analysis based on accounting career anchors using clustering techniquesPLOS ONE

Dear Dr. Avila-George,

Thank you for submitting your manuscript to PLOS ONE. After careful consideration, we feel that it has merit but does not fully meet PLOS ONE’s publication criteria as it currently stands. Therefore, we invite you to submit a revised version of the manuscript that addresses the points raised during the review process. However, I strongly encourage you to pay attention to the reviewers’ comments and recommendations, mainly and foremost, proofreading the revised version before submitting it.

We look forward to receiving your revised manuscript.

Kind regards,

Sina Safayi, D.V.M., Ph.D.

Academic Editor

PLOS ONE

Journal Requirements:

Reviewers' comments:

Reviewer's Responses to Questions

**Comments to the Author**

1. Is the manuscript technically sound, and do the data support the conclusions?

Reviewer #1: Yes

Reviewer #2: Yes

Reviewer #3: No

2. Has the statistical analysis been performed appropriately and rigorously? 

Reviewer #1: I Don't Know

Reviewer #2: I Don't Know

Reviewer #3: No

3. Have the authors made all data underlying the findings in their manuscript fully available?

Reviewer #1: Yes

Reviewer #2: Yes

Reviewer #3: No

4. Is the manuscript presented in an intelligible fashion and written in standard English?

Reviewer #1: No

Reviewer #2: No

Reviewer #3: No

5. Review Comments to the Author

Reviewer #1: The authors discuss the disconnect between what students may be taught at educational institutions vs the realities of actual on-the-job performance, worker expectations, and career anchors. The authors specifically look at the profiles of competencies characterizing the professional accountant, using data taken from Peruvian and Colombian participants.

The authors have given a good overall review of career anchor theoretical frameworks and recent literature addressing these anchors. The authors describe their method of analysis and clustering particularly well. I also enjoyed the description of the ideas articulated in the introduction and the overall concluding thoughts. However, I think the paper is not fully understandable in its current form. But it has potential to be very interesting once the authors have undertaken revision of the article, and I encourage the authors to bear in mind how interesting this work could be to non-specialists, particularly in taking greater care to explicitly point out the steps taken, and the meanings of the numbers in Table 6. This then has the potential to be much more accessible to those who work in career development spaces but do not have expertise in clustering.

Specific comments:

What exactly do the authors mean by an “academic cycle”? From Table 4, it is equated with “Level”, and refers to the students academic period - is this a standard measure of time, such as a semester? Or does this time period vary between campuses e.g. is it possible it could refer to a trimester in one and a semester in another? Making this clearer would be helpful.

The sample sizes for Peruvian vs Colombian participants are not explained, what is the reason for the discrepancy in sample size and do the authors think that anything is affected in their overall analysis by this?

How do the authors define which are the anchors that are most important based on the numerical values in Table 6? Line 211 states “whose outstanding professional anchors are entrepreneurial creativity, service, and pure challenge.” In the table, these are the 3 highest numerical values, but the number are 23, 22, 22 respectively. Autonomy, Stability and Lifestyle all have values of 21 - why are these not also important? Especially as for the 51-yer old male, values of 21 are cited (for Services) as important. The authors could explain more clearly how the numbers in Table 6 are converted into the conclusions they provide.

Given that PLOS ONE does not copyedit articles, some work needs to be undertaken to clarify the English, particularly the grammar which is understandable in the main, but not standard. For example, line 35, “do not limit the research. Such as…” should probably read as “do not limit the research, such as”. Another example is on line 39, “Include a diversity of groups, with multiple characteristics 39 [8]” does not make grammatical sense; likewise on line 73 “students who were not enrolled in the academic semester conducted the survey” presumably means “students who were not enrolled in the academic semester when the survey was conducted.” There are also other proofing errors such as “The of this research was…” on line 28, for example. Further examples: The sentence on line 218 beginning “Pure challenge anchor” does not make grammatical sense; likewise in line 224 “it can be identified a male model” does not make grammatical sense. Line 274 reads, “The explanation of this anchor is found in Section .” The name of the section being referred to is missing. The first sentence in the Conclusion does not make grammatical sense. Overall the paper was mostly understandable and I was generally able to follow the logic and arguments, but not always, and I think this affects the impact the paper could have.

One exception was the results section, where I was initially unable to follow the discussion in “Peruvian university campus: Description of cluster significance”. For example, on line 212, a sentence begins “Both clusters…”, but only one cluster has been discussed clearly so far, and so the reader cannot easily determine which two clusters are being referred to. It was only when I was reading the Colombian results that I understood that both the Peruvian clusters are what is being compared - the section does begin by saying this but it was not initially clear. It is possible that this easily could be much clearer by moving the paragraph beginning on line 224 to come before the sentence beginning “Both clusters” on line 212.

In line 232, the authors state, “Table 7 describes the Fisher statistic of 4.93, which was the highest. This indicates that among all the other anchors, this anchor has greater significance on the sample of Peruvian University students” - which anchor is “this anchor”? Is it “Creativity in Entrepreneurship”? It would be helpful to the reader to state this explicitly in the text, especially as the text that follows requires the reader have this knowledge in mind. This would also be helpful generally, to guide the reader towards the conclusions the authors are drawing throughout.

Reviewer #2: PONE-D-22-01029 titled “Competency analysis based on accounting career anchors using clustering techniques” by Avila-George et al., provides a quantitative method to analyze and compare career anchors (competencies) influencing aspirations and identity formation, by using test cases of future accountants in Peru and Colombia. The study highlights differences in values between professionals of the two countries with similar educational background, indicating influence of socio-economic cultures in career choices. The inherent merit of this study is its inclusion of diverse international perspectives and diversifying representative data in analyzing career anchors. However, the authors need to do a better job highlighting the broad significance of this study. I recommend approval with following suggestions and considerations for revisions and improvement

Major comments

• Highlight the significance and purpose of study better. It was hard to comprehend the purpose, focus and broad impact of this study until reading much of the introduction. There is no mention of the purpose or significance of the study in the abstract either. The paper is focused on methodologies far more than outlining the problem statement and the merits of investigating career anchors internationally. It’s primarily written for niche readers well versed with the field and literature (Schein and Brooks) and needs to expand its communication to include non-specialists.

• After reading the paper, I am curious whether this methodology can be used to cluster career anchors for longitudinal career progression analysis by country. For example- does career anchors and career orientations change for students (future accountants) after they spend a few years in the workforce due to influence of economic and marketplace environments?

• The authors mention skills gap and inadequate higher education training to meet the skills gaps. Can this methodology be used to define said skills gap in accounting profession by country. For example- in parallel to career orientation and anchors of students, understanding profile of career anchors and competencies by analyzing job descriptions and labor data for accountant positions in those countries will highlight clusters of shared values and disparities.

Minor comment

• The manuscript requires proofreading. There are regular typos and missing words.

Reviewer #3: There were numerous grammatical and vocabulary usage errors, which in some cases requires the reader to guess at the point being made. There were no p values reported with the ANOVA analysis. The survey questions are not available. Demographic breakdown of the survey respondents and total number of respondents is difficult if not impossible to evaluate.

6. PLOS authors have the option to publish the peer review history of their article (what does this mean?). If published, this will include your full peer review and any attached files.

Reviewer #1: **Yes: **Gary S. McDowell

Reviewer #2: No

Reviewer #3: No

---

## [Author Response · Author response to Decision Letter 0]

31 Jul 2022

%%%%%%%%%%%%%%%%

% Reviewer 1 %

%%%%%%%%%%%%%%%%

\\reviewersection

\\begin{point}

The authors discuss the disconnect between what students may be taught at educational institutions vs the realities of actual on-the-job performance, worker expectations, and career anchors. The authors specifically look at the profiles of competencies characterizing the professional accountant, using data taken from Peruvian and Colombian participants.

The authors have given a good overall review of career anchor theoretical frameworks and recent literature addressing these anchors. The authors describe their method of analysis and clustering particularly well. I also enjoyed the description of the ideas articulated in the introduction and the overall concluding thoughts. However, I think the paper is not fully understandable in its current form. But it has potential to be very interesting once the authors have undertaken revision of the article, and I encourage the authors to bear in mind how interesting this work could be to non-specialists, particularly in taking greater care to explicitly point out the steps taken, and the meanings of the numbers in Table 6. This then has the potential to be much more accessible to those who work in career development spaces but do not have expertise in clustering.

\\end{point}

\\begin{reply}

Dear reviewer, we appreciate your positive comments about our manuscript. The improvements are reported below:

\\begin{enumerate}

 \\item In the first section, Schein and Brooks' theory concerning running anchors was added to the introduction, answering the following questions: What are running anchors? What is the importance of running anchors? 

 \\item A more detailed description of the data in Table 8, which summarizes the results of our study, was added to the results section.

\\end{enumerate}

\\end{reply}

\\begin{point}

What exactly do the authors mean by an “academic cycle”? From Table 4, it is equated with “Level”, and refers to the students academic period - is this a standard measure of time, such as a semester? Or does this time period vary between campuses e.g. is it possible it could refer to a trimester in one and a semester in another? Making this clearer would be helpful..

\\end{point}

\\begin{reply}

It is the academic period of 6 months of university studies where the student develops the teaching-learning process.

\\end{reply}

\\begin{point}

The sample sizes for Peruvian vs Colombian participants are not explained, what is the reason for the discrepancy in sample size and do the authors think that anything is affected in their overall analysis by this?

\\end{point}

\\begin{reply}

The total number of accounting students in both countries was considered in both samples; this means that the number of students in Peru was higher because it had three university campuses and only one in Colombia. This is detailed in Tables 3 and 4.

\\end{reply}

\\begin{point}

How do the authors define which are the anchors that are most important based on the numerical values in Table 6? Line 211 states “whose outstanding professional anchors are entrepreneurial creativity, service, and pure challenge.” In the table, these are the 3 highest numerical values, but the number are 23, 22, 22 respectively. Autonomy, Stability and Lifestyle all have values of 21 - why are these not also important? Especially as for the 51-yer old male, values of 21 are cited (for Services) as important. The authors could explain more clearly how the numbers in Table 6 are converted into the conclusions they provide.

\\end{point}

\\begin{reply}

Considering your observation, the predominant career anchors in the manuscript's text were expanded, with a minimum value of 21 for each anchor. Therefore, the following anchors were added to the Peruvian sample: functional technician, autonomy, stability, and lifestyle. In the case of Colombia: functional technician, autonomy, and pure challenge.

\\end{reply}

\\begin{point}

Given that PLOS ONE does not copyedit articles, some work needs to be undertaken to clarify the English, particularly the grammar which is understandable in the main, but not standard. For example, line 35, “do not limit the research. Such as…” should probably read as “do not limit the research, such as”. Another example is on line 39, “Include a diversity of groups, with multiple characteristics 39 [8]” does not make grammatical sense; likewise on line 73 “students who were not enrolled in the academic semester conducted the survey” presumably means “students who were not enrolled in the academic semester when the survey was conducted.” There are also other proofing errors such as “The of this research was…” on line 28, for example. Further examples: The sentence on line 218 beginning “Pure challenge anchor” does not make grammatical sense; likewise in line 224 “it can be identified a male model” does not make grammatical sense. Line 274 reads, “The explanation of this anchor is found in Section .” The name of the section being referred to is missing. The first sentence in the Conclusion does not make grammatical sense. Overall the paper was mostly understandable and I was generally able to follow the logic and arguments, but not always, and I think this affects the impact the paper could have. One exception was the results section, where I was initially unable to follow the discussion in “Peruvian university campus: Description of cluster significance”. For example, on line 212, a sentence begins “Both clusters…”, but only one cluster has been discussed clearly so far, and so the reader cannot easily determine which two clusters are being referred to. It was only when I was reading the Colombian results that I understood that both the Peruvian clusters are what is being compared - the section does begin by saying this but it was not initially clear. It is possible that this easily could be much clearer by moving the paragraph beginning on line 224 to come before the sentence beginning “Both clusters” on line 212.

\\end{point}

\\begin{reply}

Thank you very much for your detailed comments. We have made all the suggested changes, and the manuscript has been sent to AJE for proofreading.

\\end{reply}

%%%%%%%%%%%%%%%%

% Reviewer 2 %

%%%%%%%%%%%%%%%%

\\reviewersection

\\begin{point}

PONE-D-22-01029 titled “Competency analysis based on accounting career anchors using clustering techniques” by Avila-George et al., provides a quantitative method to analyze and compare career anchors (competencies) influencing aspirations and identity formation, by using test cases of future accountants in Peru and Colombia. The study highlights differences in values between professionals of the two countries with similar educational background, indicating influence of socio-economic cultures in career choices. The inherent merit of this study is its inclusion of diverse international perspectives and diversifying representative data in analyzing career anchors. However, the authors need to do a better job highlighting the broad significance of this study. I recommend approval with following suggestions and considerations for revisions and improvement

Major comments

Highlight the significance and purpose of study better. It was hard to comprehend the purpose, focus and broad impact of this study until reading much of the introduction. There is no mention of the purpose or significance of the study in the abstract either. The paper is focused on methodologies far more than outlining the problem statement and the merits of investigating career anchors internationally. It’s primarily written for niche readers well versed with the field and literature (Schein and Brooks) and needs to expand its communication to include non-specialists.

\\end{point}

\\begin{reply}

The abstract was improved according to your suggestions; it considers the impact of the study and the significance of the proposal.

\\end{reply}

\\begin{point}

After reading the paper, I am curious whether this methodology can be used to cluster career anchors for longitudinal career progression analysis by country. For example- does career anchors and career orientations change for students (future accountants) after they spend a few years in the workforce due to influence of economic and marketplace environments?

\\end{point}

\\begin{reply}

The research is cross-sectional, but it can be applied to other samples and not necessarily to professional accountants since this survey is for all types of professionals. The predominant anchors obtained in this study should not necessarily be the same in another period since the results of this study cannot be generalized to other case studies.

\\end{reply}

\\begin{point}

The authors mention skills gap and inadequate higher education training to meet the skills gaps. Can this methodology be used to define said skills gap in accounting profession by country. For example- in parallel to career orientation and anchors of students, understanding profile of career anchors and competencies by analyzing job descriptions and labor data for accountant positions in those countries will highlight clusters of shared values and disparities.

\\end{point}

\\begin{reply}

No, this study only develops predominant anchors in a professional profile. What you can suddenly make some comparison would be of these anchors with the competencies demanded in the labor market.

\\end{reply}

\\begin{point}

The manuscript requires proofreading. There are regular typos and missing words.

\\end{point}

\\begin{reply}

The manuscript has been sent to AJE for proofreading.

\\end{reply}

%%%%%%%%%%%%%%%%

% Reviewer 3 %

%%%%%%%%%%%%%%%%

\\reviewersection

\\begin{point}

There were numerous grammatical and vocabulary usage errors, which in some cases requires the reader to guess at the point being made. 

\\end{point}

\\begin{reply}

The manuscript has been sent to AJE for proofreading.

\\end{reply}

\\begin{point}

There were no p values reported with the ANOVA analysis. 

\\end{point}

\\begin{reply}

The p-value values were added to Table 9.

\\end{reply}

\\begin{point}

The survey questions are not available.

\\end{point}

\\begin{reply}

It can be found at the following link: \\url{https://github.com/jasg1612/anchors}

\\end{reply}

\\begin{point}

The survey questions are not available.

\\end{point}

\\begin{reply}

In both samples, the total number of accounting students from both countries was considered.

\\begin{table}[ht]

\\footnotesize

\\centering

\\caption{\\bf Sample in Peru}

\\label{instrument}

\\begin{tabular}{p{0.20\\columnwidth} p{0.2\\columnwidth}p{0.2\\columnwidth}p{0.2\\columnwidth}}

\\toprule

\\textbf{Headquarter} & \\textbf{Female} & \\textbf{Male} & \\textbf{Total} \\\\

\\midrule

Juliaca & 206 & 112 & 318\\\\

Lima & 50 & 37 & 87\\\\

Tarapoto & 39 & 15 & 54\\\\

\\midrule

Total & 295 & 164 & 459\\\\

\\bottomrule

\\end{tabular}

\\end{table}

\\begin{table}[ht]

\\footnotesize

\\centering

\\caption{\\bf Sample in Colombia}

\\label{instrument}

\\begin{tabular}{p{0.20\\columnwidth} p{0.2\\columnwidth}p{0.2\\columnwidth}p{0.2\\columnwidth}}

\\toprule

\\textbf{Headquarter} & \\textbf{Female} & \\textbf{Male} & \\textbf{Total} \\\\

\\midrule

Principal & 34 & 30 & 64\\\\

\\bottomrule

\\end{tabular}

\\end{table}

\\end{reply}

---

## [Decision Letter · Decision Letter 1]

1 Sep 2022

PONE-D-22-01029R1Competency analysis based on accounting career anchors using clustering techniquesPLOS ONE

Dear Dr. Avila-George,

Thank you for submitting your manuscript to PLOS ONE. After careful consideration, we feel that it has merit but does not fully meet PLOS ONE’s publication criteria as it currently stands. Therefore, we invite you to submit a revised version of the manuscript that addresses the points raised during the review process.

Your work has the merit of getting accepted, yet it requires additional revisions and improvements. I hope you consider addressing all the comments, references, and suggestions for improvement raised by our reviewers. 

We look forward to receiving your revised manuscript.

Kind regards,

Sina Safayi, D.V.M., Ph.D.

Academic Editor

PLOS ONE

Journal Requirements:

Reviewers' comments:

Reviewer's Responses to Questions

**Comments to the Author**

1. If the authors have adequately addressed your comments raised in a previous round of review and you feel that this manuscript is now acceptable for publication, you may indicate that here to bypass the “Comments to the Author” section, enter your conflict of interest statement in the “Confidential to Editor” section, and submit your "Accept" recommendation.

Reviewer #1: (No Response)

Reviewer #3: (No Response)

Reviewer #4: All comments have been addressed

2. Is the manuscript technically sound, and do the data support the conclusions?

Reviewer #1: Yes

Reviewer #3: No

Reviewer #4: (No Response)

3. Has the statistical analysis been performed appropriately and rigorously? 

Reviewer #1: I Don't Know

Reviewer #3: I Don't Know

Reviewer #4: (No Response)

4. Have the authors made all data underlying the findings in their manuscript fully available?

Reviewer #1: Yes

Reviewer #3: Yes

Reviewer #4: (No Response)

5. Is the manuscript presented in an intelligible fashion and written in standard English?

Reviewer #1: Yes

Reviewer #3: Yes

Reviewer #4: (No Response)

6. Review Comments to the Author

Reviewer #1: First of all, many thanks to the authors for the changes they have made, which have greatly improved the clarity of the manuscript. I still have some minor outstanding comments:

The clusters in Table 8 show that for Peru, both clusters are represented by the same gender (I am guessing male, but it isn't clear because the scoring isn't defined). Can the authors describe what this means? What "happens" to females in the sample?

Another interesting aspect that is apparent from Figures 4 and 5 and Table 8 is that one cluster has scores that are all lower for the anchors than the other cluster. Is this expected? Would it not be possible that clusters would emerge with each having anchors higher than the other cluster? It would be helpful to see this approached in the discussion.

It is still not clear what is meant by an "academic cycle" in the text of the paper - this needs to be explained thoroughly. In the response to my previous review, the authors replied to me that, "It is the academic period of 6 months of university studies where the student develops the teaching-learning process." I'm still not clear what this means; how many 6 month periods are there in a student's course? Are there 2 periods in a year, or does each cycle represent a new year? Perhaps there needs to be a short explanation of the structure of the accountancy courses in the introduction to make this clearer. Another alternative that the authors could consider - are the cycles relevant to their conclusions? The cycles are only mentioned as an additional characteristic but I don't think there is any discussion of the significance. Are later (e.g. 4/5) more important? Is there a distribution of recipients across cycles? If it's not important to the author's conclusions, perhaps all mention of cycles could just be removed. Either way, I am still confused by the concept and so am concerned many readers will be too.

One minor edit needed is that the Figures are not comprehensively referenced in the text; for example, it is not trivial to find the text that aligns with Figure 4 in order to appreciate the context of the Figure. Figure 5 is mentioned however.

Another minor edit related to the figures is to add text to the figure legends to clarify definitions/colors in used. One example: which colors in Figures 4 and 5 refer to the anchor? What do the diamonds denote? Please include explanations in the figure legends. Another example: in the figure legend for Table 8, it should be explained clearly what '1' means for gender, and what '4' means for academic level. While it's possible to figure this out from the main text, the figure legend should contain all of the information so that anyone can look at the figure, read the legend, and understand all of the information contained within it, without having to hunt through the text to find an explanation. I think the significance of the author's work will be much more readily apparent with greater clarity about what the figures describe in the text of the figure legends. Lines 275 to 286 are an example of where this work happens in the main text; but the figure legend should also allow the reader to come to this conclusion themselves, so that they are able to agree with the authors' conclusion in lines 275 to 286.

Once more, with relation to the figures - the y axes should be defined to describe what the numbers signify. And the y axes in Figures 4 and 5 differ within the figure for each cluster; they should be the same in order to allow the comparison that the figure begs.

Lines 212 and 302 - The name of the Section in which the explanation can be found is missing.

Reviewer #3: The authors have done a commendable job of responding to the reviewers’ points, and I agree that there is value in comparing the motivations of students for a career in accounting between different cultures and countries. However, I have serious concerns about the paper and do not believe it could be ready for publication without major revisions.

The aim of the study as defined in the abstract and introduction is not consistent with the chosen survey instrument. The abstract states that “This study aims to define the career anchors of accounting students and the skills and knowledge required to be learned during professional training.” The introduction explains the aim like this: “The aim of this research was to analyze the profile of competencies that characterize professional accountants.” Although the career anchors are discussed in detail, there is no mention in the paper of skills development, training program requirements, or competencies.

In fact, the 40-question survey, the only source of data for the study, is a commonly used survey of workplace values and preferences. The survey is available in Spanish here: https://github.com/jasg1612/anchors/blob/main/cuestionario.docx and, translated to English, the survey questions read:

1 I would like to be so good at what I do that people continually ask me for advice and suggestions.

2 I am more satisfied with my work when I can integrate and manage the efforts of others

3 I would like to have a career that allows me autonomy and decide the deadlines

4 Security and stability are more important to me than freedom and autonomy

5 I am always looking for ideas that allow me to have my own business

6 I consider that I achieve success in my career only if I have the feeling of having contributed to the common good

7 I would like a career where I can solve problems or come out on top in very challenging situations

8 I would rather leave my company than occupy a position that would compromise my attention to my family and personal life

9 For me, success consists of developing my technical or functional abilities until I become an expert

10 I would like to be in charge of a complex organization and make decisions that affect many people

11 I am more satisfied when I have complete freedom to define my own activities, deadlines and procedures

12 I would rather leave my company than accept a project that would affect my security within the organization

13 Starting my own business is more important than reaching a senior management position in another organization

14 I am more satisfied with my career when I can put my talent at the service of others

15 I achieve success in my career only if I face and overcome great challenges and challenges

16 I would like a career that allows me to integrate my personal, family and professional needs

17 I am more attracted to becoming a senior manager within my functional area than becoming CEO

18 I achieve success in my career only if I become CEO of a company

19 I achieve success in my career only if I achieve autonomy and full freedom

20 I seek work within organizations that provide me with security and stability

21 I am more satisfied with my career when I have created something that is the result of my own ideas and efforts

22. It is more important to me to use my abilities to create a world where people live and work better than to have a high-level managerial position.

23. I have found myself more satisfied in my career when I have solved seemingly insoluble problems or won when it seemed impossible to do so.

24 I am satisfied with my life only when I manage to achieve a balance between the demands of my personal, family and professional life

25 I would rather leave my company than accept a project that would force me to leave my area of specialization

26 I am more attracted to becoming a CEO than a senior manager within my area of expertise

27 The opportunity to do a job according to my own criteria, without rules and limitations, is more important to me than safety

28 I feel more satisfied with my job when I consider that I have achieved financial and professional security

29. I consider that I achieve success in my career only if I manage to create or build something that is completely my own product or idea.

30 I would like to have a career that makes a great contribution to humanity and society

31 I look for job opportunities that test my ability to solve problems or to compete

32 Finding a balance between the demands of my personal and professional life is more important than landing a high-level managerial position

33 I am more satisfied with my work when I have the opportunity to use my skills and talents

34 I would rather leave my company than accept a position that takes me away from the path to general management

35 I would rather leave my company than accept a position that limits my autonomy and freedom

36 I would like to have a career that allows me to feel a certain level of security and stability

37 I would like to create and build my own business

38 I would rather leave my company than accept a project that imitated my ability to help others

39 Working on seemingly insoluble problems is more important than reaching a high-level managerial position

40 I have always looked for professional opportunities that do not interfere too much with my personal and family concerns

These questions are integral to understanding a trainee’s workplace preferences and long term career goals, and the survey is a important tool used by career counselors, but, it is my opinion that this survey, in and of itself, does not represent a novel data collection tool that could form the basis of a peer reviewed paper. Nor do the questions allow any conclusions about training competencies or accounting-specific knowledge.

To advance the field of accountant training and career development it is important to present data that will inform suggestions about program graduates’ career readiness, knowledge acquisition, employment outcomes, performance, etc.

If the purpose of the paper is to show how clustering algorithms can help researchers to better understand survey responses, then this should be stated as the aim of the study. And if this is to be the case, the rationale for the methods used needs to be explained more thoroughly. For example, using three clustering methods (KMEANS, DBSCAN, and BIRCH) without an a priori rational and then stating simply that DBSCAN was chosen because it “performs best” (line 216) with no further explanation leads this reader to conclude that the authors have not made themselves experts in clustering methods.

Another aspect of the paper that is not clear is why there are only two clusters assigned? How were those clusters defined? Can average scores for each career anchor be reported for each cluster? Perhaps this is what table 8 is attempting to describe, but the columns for the career anchors are described in the methods section as being the sum of all group records (line 222), but the numbers are all less than 30 when hundreds of surveys were recorded. Similarly, reporting averages for each career anchor in each cluster would be more valuable than choosing one representative individual in each cluster and reporting about their survey responses as is done in the lines 266-274.

Table 9 is also not well-described in the methods section or the results section. The authors use the data in Table 9 to draw a conclusion about the imagination and innovative abilities of the survey respondents (line 257), but as stated above, the survey does not address competencies, skills, or proficiency from a self reported lens, and definitely not from an impartial external lens.

Other elements of the paper that would need to be fixed include multiple instances where sections are referred to without section numbers (e.g. Line 212, 268, 302). The survey response rate for both locations should be reported. Table 5 contains multiples of question numbers in the same anchor categories.

Reviewer #4: (No Response)

7. PLOS authors have the option to publish the peer review history of their article (what does this mean?). If published, this will include your full peer review and any attached files.

Reviewer #1: **Yes: **Gary S. McDowell

Reviewer #3: No

Reviewer #4: No

---

## [Author Response · Author response to Decision Letter 1]

16 Nov 2022

Reviewer 1

1.1 — The clusters in Table 8 show that for Peru, both clusters are represented by the same gender

(I am guessing male, but it isn’t clear because the scoring isn’t defined). Can the authors describe what this means? What ”happens” to females in the sample?

Reply: Derived from their punctual comments, the experimentation has been redefined with the search for the best hyperparameters used by the algorithms to cluster the data. The algorithms were compared using the Silhouette coefficient metric, which measures the level of cohesion of the clustered data. Thus defining the best clustering algorithm. The result of this comparison was that the DBSCAN algorithm obtained the best measure of cohesion, as described in Table 8, in both samples from Peru and Colombia. Once the DBSCAN algorithm labeled the datasets’ rows, each cluster’s mean was calculated. Table 9 shows the summary of the labeling of the data from cluster 0 of each of the two datasets; the data from cluster -1 for each dataset were discarded because the number of records was not representative for calculating a mean (i.e., these data could be considered as noise), see Figures 4 and 5. Clustering in the present work was used for the dimensional reduction of the data, in this case, applied to reduce the number of career anchors to the most prevalent anchors in each sample of Peru and Colombia. The latter process was validated by the Fisher’s F statistic of ANOVA, which measures the correlations between the career anchors with cluster 0, identifying by the estimated value those with the highest value that was prevalent in the clusters. On the other hand, the female gender does not appear in any of the clusters because the male gender was prevalent among the respondents.

1.2 — Another interesting aspect that is apparent from Figures 4 and 5 and Table 8 is that one cluster has scores that are all lower for the anchors than the other cluster. Is this expected? Would it not be possible that clusters would emerge with each having anchors higher than the other cluster? It would be helpful to see this approached in the discussion.

Reply: The scores of the anchors will depend on the values of the group of records labeled by the cluster, since the anchors were averaged and this measure is susceptible to the values of the anchors at the time of averaging. The DBSCAN algorithm detected cluster 0 and -1. Cluster -1 had a minimum number of records belonging to the cluster, therefore this cluster is discarded, leaving only the only cluster 0, observing the prevalences in this cluster 0. The idea of the research was the dimensional reduction of the career anchors therefore a representative cluster was expected in the whole sample defining the prevalent anchors in each sample, by the mean scores defined in each anchor.

1.3 — It is still not clear what is meant by an ”academic cycle” in the text of the paper - this needs to be explained thoroughly. In response to my previous review, the authors replied to me that, ”It is the academic period of 6 months of university studies where the student develops the teaching-learning process.” I’m still not clear what this means; how many 6-month periods are there in a student’s course? Are there 2 periods in a year, or does each cycle represent a new year? Perhaps there needs to be a short explanation of the structure of the accountancy courses in the introduction to make this clearer. Another alternative that the authors could consider - are the cycles relevant to their conclusions? The cycles are only mentioned as an additional characteristic but I don’t think there is any discussion of the significance. Are later (e.g. 4/5) more important? Is there a distribution of recipients across cycles? If it’s not important to the author’s conclusions, perhaps all mention of cycles could just be removed. Either way, I am still confused by the concept and so am concerned many readers will be too. 

Reply: We understand your doubts, you are right that the way the article is written causes confusion. Therefore, it was considered pertinent to eliminate the term academic cycle, since the anchors are part of the academic formation of the students throughout their stay at the university.

1.4 — One minor edit needed is that the Figures are not comprehensively referenced in the text; for example, it is not trivial to find the text that aligns with Figure 4 in order to appreciate the context of the Figure. Figure 5 is mentioned however.

Reply: The figures were improved according to the explanation in Table 9; the boxplots describe the data interval of each cluster and the mean described in the table. The reference to figure 4 was added on line 234, its explanation

1.5 — Another minor edit related to the figures is to add text to the figure legends to clarify definitions/colors in used. One example: which colors in Figures 4 and 5 refer to the anchor? What do the diamonds denote? Please include explanations in the figure legends. Another example: in the figure legend for Table 8, it should be explained clearly what ’1’ means for gender, and what ’4’ means for academic level. While it’s possible to figure this out from the main text, the figure legend should contain all of the information so that anyone can look at the figure, read the legend, and understand all of the information contained within it, without having to hunt through the text to find an explanation. I think the significance of the author’s work will be much more readily apparent with greater clarity about what the figures describe in the text of the figure legends. Lines 275 to 286 are an example of where this work happens in the main text; but the figure legend should also allow the reader to come to this conclusion themselves, so that they are able to agree with the authors’ conclusion in lines 275 to 286.

Reply: In response to their recommendations, the legend to Table 09 has been improved, explaining the numbers appearing in the age column, and a legend has been added to Figures 4 and 5, explaining the boxplot colors, the points outside the interval.

1.6 — Once more, with relation to the figures - the y axes should be defined to describe what the numbers signify. And the y axes in Figures 4 and 5 differ within the figure for each cluster; they should be the same in order to allow the comparison that the figure begs.

Reply: The Y-axis values in Figures 4 and 5 have been standardized; a legend has been added to both Figures explaining the meaning of the Y-axis values.

1.7 — Lines 212 and 302 - The name of the Section in which the explanation can be found is missing.

Reply: The cross-reference has been corrected in both sections.

REVIEWER 2

2.1 — The aim of the study as defined in the abstract and introduction is not consistent with the chosen survey instrument. The abstract states that “This study aims to define the career anchors of accounting students and the skills and knowledge required to be learned during professional training.” The introduction explains the aim like this: “The aim of this research was to analyze the profile of competencies that characterize professional accountants.” Although the career anchors are discussed in detail, there is no mention in the paper of skills development, training program requirements, or competencies.

Reply: The wording of the objective presented in the introduction has been improved to be consistent with the study conducted and all sections of the paper.

This research work aims to identify the prevalent anchors in the professional accounting career using the Schein scale and to describe the prevalent anchors by defining the values, attitudes, aptitudes, skills, and interests and represent core elements in forming professionals.

2.2 — These questions are integral to understanding a trainee’s workplace preferences and long term career goals, and the survey is a important tool used by career counselors, but, it is my opinion that this survey, in and of itself, does not represent a novel data collection tool that could form the basis of a peer reviewed paper. Nor do the questions allow any conclusions about training competencies or accounting-specific knowledge.

Reply: The instrument is based on Jos ´e Medina’s adaptation of Schein’s anchor theory, directing each question to each concept and purpose of the anchor. The author of the book: Lead your career: Don’t let others decide for you has extensive experience in personnel selection and research in companies, has a doctorate in psychology, and is an expert in Organization Development from the National Training Laboratories Institute in Washington D.C. This shows us that the instrument has a theoretical foundation on the side of Schein’s anchors, it is based on the judgment of the expert Jos ´e Medina. Crombach’s alpha statistical reliability test was also performed with a result of 0.94 for the instrument. In this sense, the study’s purpose is to understand students’ workplace preferences and long-term career goals.

2.3 — To advance the field of accountant training and career development it is important to present data that will inform suggestions about program graduates’ career readiness, knowledge acquisition, employment outcomes, performance, etc. If the purpose of the paper is to show how clustering algorithms can help researchers to better understand survey responses, then this should be stated as the aim of the study. And if this is to be the case, the rationale for the methods used needs to be explained more thoroughly. For example, using three clustering methods (KMEANS, DBSCAN, and BIRCH) without an a priori rational and then stating simply that DBSCAN was chosen because it “performs best” (line 216) with no further explanation leads this reader to conclude that the authors have not made themselves experts in clustering methods.

Reply: Thepurposesofthestudywereexplainedintheintroduction,theabstractandintheexploratory analysis section. To achieve this purpose, the explanatory research was used, which is based on the dimensional reduction of the variables that explain the case study, which consists of the career anchors of professionals in accounting, by reducing the dimensions, the prevalent anchors in the student’s training were found, being able to make the corresponding analysis then of the skills found, aptitudes, attitudes developed in such training. After performing the dimensional reduction of the anchors with the clustering, a second statistical test was performed to cross the information of the clusters with the ANOVA test and to explain the prevalent anchors.

2.4 — Another aspect of the paper that is not clear is why there are only two clusters assigned? How were those clusters defined? Can average scores for each career anchor be reported for each cluster? Perhaps this is what Table 8 is attempting to describe, but the columns for the career anchors are described in the methods section as being the sum of all group records (line 222), but the numbers are all less than 30 when hundreds of surveys were recorded. Similarly, reporting averages for each career anchor in each cluster would be more valuable than choosing one representative individual in each cluster and reporting about their survey responses as is done in the lines 266- 274.

Reply: Another aspect of the document that is not clear is why only two clusters are assigned? How were these clusters defined? Hyperparameter tuning, described in Table 7, was performed using the cohesion metric, silhouette, with the best algorithm determining the number of clusters using the optimized hyperparameters Can the mean scores of each run anchor be reported for each of each cluster? The values that were averaged were those assigned in each anchor, product of the sum of the questions related to each anchor then were these values averaged taking into consideration the cluster that was labeled with anchor 0 and anchor -1, the records were eliminated for representing a small amount. This average value was below the value of 30, since most of the values were below 30.

2.5 — Table 9 is also not well-described in the methods section or the results section. The authors use the data in Table 9 to draw a conclusion about the imagination and innovative abilities of the survey respondents (line 257), but as stated above, the survey does not address competencies, skills, or proficiency from a self-reported lens, and definitely not from an impartial external lens.

Reply: Clustering was used to reduce the dimensions of the data; in this case, it was applied to find the most prevalent anchors in each sample from Peru and Colombia (the purpose of the study). This process was validated by the Fisher’s F statistic of ANOVA, which measures the correlations between the anchors run with cluster 0, identifying by Fisher’s estimated value those with the highest prevalence value in the cluster. In the methodology, the exploratory analysis and statistical test section explain how ANOVA works with clustering as a statistical tool to confirm the dimensional reduction analysis of clustering.

2.6 — Other elements of the paper that would need to be fixed include multiple instances where sections are referred to without section numbers (e.g. Line 212, 268, 302). The survey response rate for both locations should be reported. Table 5 contains multiples of question numbers in the same anchor categories.

Reply: Added cross references to lines 212, 268, 302 and corrected items in Table 5.

---

## [Decision Letter · Decision Letter 2]

20 Dec 2022

Competency analysis based on accounting career anchors using clustering techniques

PONE-D-22-01029R2

Dear Dr. Avila-George,

We’re pleased to inform you that your manuscript has been judged scientifically suitable for publication and will be formally accepted for publication once it meets all outstanding technical requirements.

Kind regards,

Sina Safayi, D.V.M., Ph.D.

Academic Editor

PLOS ONE

Additional Editor Comments (optional):

Reviewers' comments:

Reviewer's Responses to Questions

**Comments to the Author**

1. If the authors have adequately addressed your comments raised in a previous round of review and you feel that this manuscript is now acceptable for publication, you may indicate that here to bypass the “Comments to the Author” section, enter your conflict of interest statement in the “Confidential to Editor” section, and submit your "Accept" recommendation.

Reviewer #1: (No Response)

Reviewer #3: All comments have been addressed

Reviewer #4: (No Response)

2. Is the manuscript technically sound, and do the data support the conclusions?

Reviewer #1: Yes

Reviewer #3: Yes

Reviewer #4: (No Response)

3. Has the statistical analysis been performed appropriately and rigorously? 

Reviewer #1: I Don't Know

Reviewer #3: Yes

Reviewer #4: (No Response)

4. Have the authors made all data underlying the findings in their manuscript fully available?

Reviewer #1: Yes

Reviewer #3: Yes

Reviewer #4: (No Response)

5. Is the manuscript presented in an intelligible fashion and written in standard English?

Reviewer #1: Yes

Reviewer #3: Yes

Reviewer #4: (No Response)

6. Review Comments to the Author

Reviewer #1: In this study, the authors seek to identify and describe the career anchors that guide university students undertaking education towards a professional accounting career. The manuscript has been improved significantly and is much clearer, and I have only minor comments.

Minor Points:

1. Generally there are some points of clarity in language to be ironed out e.g. the first sentence in the abstract should probably end with “of university-level accountancy students” or similar; line 194 should read “are explained”; etc.

2. I just want to ask the authors to double-check that on line 62 they do mean that all students in each country are of that country’s nationality i.e. that the students at the Peruvian institutions are all Peruvian, and those at the Colombian institution are all Colombian, as they imply, and there are no international students included (if there are, that’s not a problem, it should just be articulated as e.g. “respondents from Peruvian institutions”).

Reviewer #3: I have read through the authors’ latest manuscript and feel that they have addressed all the points raised in my previous reviews. I feel that the paper is ready for publication in PLOS One.

Reviewer #4: (No Response)

7. PLOS authors have the option to publish the peer review history of their article (what does this mean?). If published, this will include your full peer review and any attached files.

Reviewer #1: **Yes: **Gary McDowell

Reviewer #3: No

Reviewer #4: No

---

## [Editor Report · Acceptance letter]

27 Dec 2022

PONE-D-22-01029R2 

Competency analysis based on accounting career anchors using clustering techniques 

Dear Dr. Avila-George:

I'm pleased to inform you that your manuscript has been deemed suitable for publication in PLOS ONE. Congratulations! Your manuscript is now with our production department. 

Kind regards, 

on behalf of

Dr. Sina Safayi 

Academic Editor

PLOS ONE